# What you say and what I hear—Investigating differences in the perception of the severity of psychological and physical violence in intimate partner relationships

**Sverker Sikström**[1]*, **Mats Dahl**[1], **Hannah Lettmann**[1], **Anna Alexandersson**[1], **Elena Schwörer**[1], **Lotta Stille**[1], **Oscar Kjell**[1], **Åse Innes-Ker**[1], **Leonard Ngaosuvan**[2]

**1** Department of Psychology, Lund University, Lund, Sweden, **2** Department of Culture and Society, Linköping University, Linköping, Sweden

* sverker.sikstrom@psy.lu.se

**Data Availability Statement:** The data can be found at OSFHOME https://osf.io/6sedp/.

## Abstract

The correct communication of the severity of violence is essential in the context of legal trials, custody cases, support of victims, etc., for providing fair treatment. A narrator that communicates their experiences of interpersonal violence may rate the seriousness of the incident differently than a rater reading the narrator's text, suggesting that there exist *perceptual differences* (PD) in severity ratings between the narrator and the rater. We propose that these perceptual differences may depend on whether the narrative is based on physical or psychological violence, and on gender differences. Physical violence may be evaluated as more serious by the receiver of the narrative than by the narrator (*Calibration PD*), whereas the seriousness of psychological violence may be difficult to convey, leading to a discrepancy in the seriousness ratings between the narrator and the rater (*Accuracy PD*). In addition, gender stereotypes may influence the seriousness rating (*Gender PD*), resulting in violence against women being perceived as more serious than the same violence against men. These perceptual differences were investigated in 3 phases using a new experimental procedure. In Phase 1, 113 narrators provided descriptions and seriousness ratings of self-experienced physical and psychological violence in relationships. In Phase 2, 340 independent raters rated the seriousness of 10 randomly selected narrations from Phase 1. In Phase 3, the genders in the narrations were changed to the opposite gender, and seriousness ratings were collected from 340 different raters. Our results confirmed the hypothesized perceptual differences. Violence to male victims was considerably more likely to be seen as severe when the raters were misled to believe the victim was a woman. We propose that these data provide practical guidelines for how to deal with misinformation in the communication of violence. The data also show that mean values and the confidence of such severity ratings need to be adjusted for several factors, such as whether it is self-experienced or communicated, the type of violence, and the gender of the victims and raters.

**Funding:** This research was supported by grants from Vinnova, 2018-02007, www.vinnova.se (SS), Kamprad Foundation. ref # 20180281, https://familjenkampradsstiftelse.se/ (SS), The Swedish Research Council. www.vr.se 2015-01229 (SS).

**Competing interests:** The authors have declared that no competing interests exist.

# Introduction

## Physical and psychological intimate partner violence

The correct evaluation of severity of violence is crucial in several contexts. An incorrect, or poor, evaluation of violence, could have legal implications. An innocent person may be charged for an offence that he/she did not commit. Guilty offenders may be liberated from sentencing for a violent act that he/she committed. The correct evaluation of violence could also have important implications in custody cases, leading to the unjustified separation of a child from a parent, or children being harmed. The aim of this paper is to investigate possible errors in communication of the severity of violence within intimate partners relations.

Intimate Partner Violence (IPV) has been shown to be a commonly underestimated problem [1] causing serious health problems among both male and female victims in societies around the world (e.g., [2–4]). The World Health Organization (WHO) defines IPV as "any behavior within an intimate relationship that causes physical, psychological, or sexual harm to those in the relationship" [5]. The intimate partner can be anything from a dating partner to a spouse, and refers to both current and former relationships. Examples of physical violence are slapping, hitting, kicking, and beating, while examples of psychological violence are humiliation, threats, and controlling behaviors, such as isolation from family or monitoring movements [5].

Physical violence is perhaps the most commonly studied type of violence [6]. Here, men's physical violence against women has predominantly been studied, whereas less research has focused on women's violence against men and violence in same-sex relationships [7–9]. However, Nybergh, Taft, Enander and Krantz [10] demonstrated in a Swedish population that exposure to violence in intimate partner relationships is not only common, but roughly equally frequent among men and women.

Psychological violence has received less attention than physical violence. Here, a complicating issue is the lack of consistent definitions [11–13] or consensus about psychological violence [14]. This may lead to poor understanding and identification of victims of psychological violence, as well as providing an erroneous background for evaluations of legal consequences. Even if psychological violence is less visible, it may have more serious consequences than physical violence, resulting in physical and mental health problems [15].

According to the latest self-report measures published by the Office for National Statistics [16], 4.2% of the population (aged 16 to 74 years) experienced domestic abuse by a partner in the UK during 2018. Most of these victims were women. The World Health Organization [5] states that the UK lifetime prevalence for sexual violence by a partner was 16%, the lifetime prevalence for physical abuse by partner was 25% and the lifetime prevalence for psychological violence as high as 34%.

Johnson [18] suggested that IPV should be divided into situational couple violence, intimate terrorism, violent resistance, and mutual violent control. Situational couple violence occurs when verbal disagreements are transitioned into physical expressions and consist of mild physical attacks such as throwing objects and slaps in the face. It is driven by temporal emotional outbursts of displeasure or disappointment and rarely inflicts serious physical harm. In contrast, intimate terrorism is driven by a need for control over the partner, and it leads to threats, coercive behaviors, obsessive surveillance, or physical attacks. Violent resistance is self-defense from intimate terrorism. Finally, mutual violent control is when a couple can be described as "two intimate terrorists battling for control".

## Gender differences

IPV is obviously closely connected to gender differences, since most partner relationships are heterosexual. The latest statistics from the Office for National Statistic [16] showed that

amongst the 2.4 million adults that experienced domestic abuse in 2018, 1.6 million were women and 786,000 were men. According to the Centers for Disease Control and Prevention [17], about 41% of the female IPV survivors experienced some form of physical injury, whereas 14% of male victims were injured. The statistics also show that women were more often than men subjected to psychological and sexual violence [16]. Johnson's [18] taxonomy explains some selective results from gender comparisons, where some [19–21] suggest that women and men are equally victimized by IPV, while others [22] report that women are more victimized. The taxonomy elegantly explains this, as situational couple violence is relatively gender equal, but women suffer more from intimate terrorism. For instance, women experience more severe violence [22], more coercive control [23], more overlapping forms of violence [24], and are more likely victims of sexual violence than men [23, 24].

Although violence against women is recognized as a global problem, women are not always the victims of IPV. Some studies have found that women are just as likely as men to inflict IPV [19–21], as both men and women may resort to violence to resolve conflicts in an intimate relationship [25]. Cho [26] even found that women inflicted IPV more frequently than men, and that they initiated physical arguments more often than men. The results from these studies are in line with statistics from institutions offering support for victims of IPV. They have encountered an increasing number of female perpetrators and male victims since countries such as the US have adopted so-called 'zero-tolerance' policies [27]. However, the number of male victims of IPV might still be underreported, as victimization by a female partner is considered emasculating and therefore highly stigmatized. This complicates identification and targeted treatment for male victims [28].

## Perception and evaluation of intimate partner violence

Although a considerable amount of prior research has studied violence in relation to gender and types of violence (e.g., physical or psychological), less research has been made into how severity of violence is communicated and influenced by perceptual differences. That is, how do victims communicate the violence they experience, and how is this narrative received and rated for severity by another person.

Although not tested directly, a literature review suggests that psychological violence is perceived as more harmful than physical violence by the victims, whereas physical violence is considered more harmful by outside observers [29]. For example, Follingstad, Rutledge, Berg, Hause and Polek [30] found that 72% of women who experienced physical and psychological violence reported the latter as more harmful. Capezza and Arriaga [2] found that outside observers reading hypothetical conflict scenarios evaluated even mild forms of physical violence as more serious than any level of psychological violence. Also, a qualitative study that focused on groups of older women showed that nonphysical abuse might be more difficult to endure and have more lasting effects than physical violence [31]. Similar results were found in a self-report study of 103 married couples [32].

The perception and evaluation of IPV is influenced by gender. For example, the ability to detect psychological violence has been found to be generally lower in men than in women [33]. A possible interpretation is that men are simply less affected by this form of violence, and consequently, report it as less serious compared to women experiencing the same form of abuse. Additionally, there may be a gender difference in how violence is remembered. Men tend to perceive physical threats from women as less serious than threats from other men [10]. Consequently, they may be less likely to remember it due to its reduced salience. In contrast, women break gender stereotypes by using physical violence, thereby increasing the likelihood of remembering it themselves [10].

Regarding the evaluation of IPV, female-perpetrated violence is less often recognized as IPV in contrast to male-perpetrated violence [34]. Compared to female perpetrators, male

perpetrators were viewed as more serious, their behavior was more likely considered illegal, and they were considered more likely to repeat violence [9, 35]. These findings indicate a predominant perception of males as the typical perpetrators of violence. In turn, this leads to a commonly accepted disapproval of men's status as victims of IPV [36]. Consequently, men minimize more and seek help less often when feeling abused [34], which can be traced back to the perceived violation of stereotypical gender roles [33]. In conclusion, men are less likely than women to view aggression as a crime and thus may be less willing to report it [37].

## The current study: Perceptual differences in communication of seriousness of violence

It is important to identify and eliminate discrepancies between the sender's intention when communicating a message, and how the message is evaluated by the receiver, in order to achieve a legally secure process. This study focuses on measuring perceptual differences in the communication and perception of violence. Communication of violence includes an experiencer (the narrator) involved in at least one violent event, who communicates this/these event(s) to another person (the rater). We are particularly interested in measuring one aspect of this communication, namely, the seriousness of the violence. Seriousness was operationalized as responses recorded on a rating scale by the narrator and the person to whom the event was communicated (rater). However, ratings were not communicated between them. Here we introduce the concept of perceptual difference (PD), which we define as the difference in severity ratings between the narrator and the rater. These perceptual differences can be further divided into three different types as outlined below. To our knowledge, these perceptual differences have not been explored in prior research using our experimental procedure.

The *Calibration PD* means that there is a significant difference between the mean seriousness rating of the narrator (i.e., the person who experienced the violent event) and that of the rater (i.e., the person that reads about the violent event) due to a systematic error. For example, if the average seriousness rating of psychological violence is rated as eight (on a scale from 0 to 10) by the narrator, and the rater rates it as five, then the difference between the average ratings of these two groups constitutes a Calibration PD of three.

The *Accuracy PD* means that the experiencers' seriousness rating of an IPV event is poorly predicted by the rater reading the narration of this event. Accuracy PD can be measured by correlating the seriousness ratings of the experiencer and the rater. A low correlation (e.g., $r = 0.3$) would indicate an Accuracy PD, whereas a high correlation (e.g., r = 0.9) would indicate no, or little, Accuracy PD.

Notice that the Calibration and Accuracy PDs can be independent of each other in the sense that it is statistically possible to have a Calibration PD without an Accuracy PD, or vice versa, as illustrated in Fig 1. For example, the mean value of the experiencers and raters ratings may agree (i.e. no Calibration PD) although the correlation is low (i.e. an Accuracy PD) as can be seen in the lower left table. Alternatively, the mean ratings may differ (i.e. a Calibration PD) in combination with a high correlation (i.e., no Accuracy PD) as visualized in the upper right panel.

Finally, a *Gender PD* exists if the seriousness ratings of violence depend on the gender of experiencers and/or the raters.

## Theoretical view and hypotheses for perceptual differences in the communication of physical and psychological violence

We suggest that it is more difficult to communicate psychological violence than physical violence. It is argued that this occurs because psychological violence works as an indirect reinforcer, meaning that psychological violence requires a learning process. This does not apply to

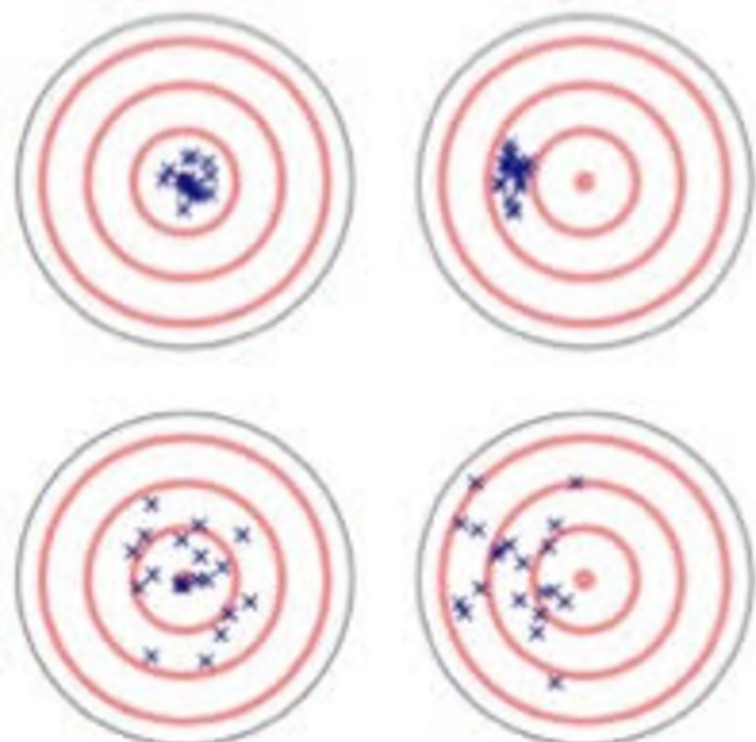

**Fig 1. A conceptual illustration of the Calibration and Accuracy PDs.** *Note.* The upper-left panel illustrates an efficient communication of severity of violence (i.e., low Calibration PD and low Accuracy PD) whereas the lower-right panel poor communication (i.e., high Calibration PD and high Accuracy PD). The upper-right panel shows poor calibration but good accuracy, whereas the lower-left panel good calibration but poor accuracy.

physical violence because it is a direct reinforcer. A direct reinforcer (e.g., a slap) produces an immediate unconditional and negative response (pain). An indirect reinforcer (e.g., a condescending statement such as 'you are worthless') produces a negative response (e.g., a feeling of worthlessness), given that the association between a conditional stimulus (e.g., 'you are worthless') and a negative outcome (e.g., the feeling of worthlessness) has previously been learned. Various forms of social learning are, of course, involved in both physical violence and psychological violence. Their difference, however, is how the forms of violence are carried out. The effect of a kick or a blow needs very little or no interpretation, since the pain it inflicts is unconditional and direct. The situation, however, needs to be interpreted, i.e., why the kick was delivered. This also holds true for psychological violence, when, for example, a threat in a given situation needs to be interpreted. Contrary to a kick or a slap, however, the statement constituting the spoken threat must be understood and interpreted in itself, before it can be identified as a threat. Thus, statements about psychological violence permit a greater variation of interpretation across situations compared to physical violence. For instance, sarcasm and irony make it more difficult for the receiver to evaluate the severity of violence in communicated statements. This leads to lower rates of agreement among raters' evaluations of the seriousness of psychological violence. We argue that this view has two implications. First, the difficulty of communicating psychological violence may lead to a Calibration PD where psychological violence is perceived as less serious when it is communicated (H1). Second, an Accuracy PD is hypothesized where psychological violence, being an indirect reinforcer, is more difficult to communicate than physical violence (i.e., lower correlations between

narrators' and raters' ratings). Consequently, the agreement in seriousness ratings between narrators and raters should be higher for narrations about physical violence than for narrations about psychological violence (H2).

Finally, we assume that gender stereotypes may influence the perception of violence (Gender PD). In particular, we hypothesize that physical violence against women is rated as more serious by both men and women (H3), due to females' comparative lack of physical strength. Consequently, they may be seen as unable to defend themselves against male perpetrators of violence.

## Method and results

### Data collection

The study consists of three phases. In the first phase, we collected narratives and ratings of experienced physical and psychological IPV incidents. In the second phase, these narratives were read and rated by independent raters. In the third phase, the gender of the collected narratives was switched, and a new set of raters read and rated the narrations. The results were analyzed with R, JASP and SPSS, where the choice of software depended on the type of analysis that was required.

### Ethics

Ethical approval for this study was granted by Lund's Regional Ethical Review Board, and adheres to their guidelines (EPN 2015/53). Participants gave written approval of their voluntarily participation in the study.

**Phase 1: Collection of narrations.** *Participants*. Data for this study was collected through Prolific Academic (https://prolific.ac/demographics), an online tool for recruiting participants that is based in the United Kingdom, but includes participants throughout the world. Using this tool, we selected participants from the USA. Participants provided informed consent, and were told that they participated voluntarily and could withdraw from participating at any time without justification. Participants were pre-screened for nationality (US), first language (English), and sexual orientation (heterosexuality). Cultural differences among participants were minimized by selecting a population from one country, and we chose a US population because the Prolific Academic site has a large number of US participants (but few, e.g., Swedish participants). The screening of heterosexuality was conducted because we were interested in focusing on hetrosexual couples, the most common sexual category. Furthermore, we did not collect data on the race of the participants, nor did we explicitly ask if the participants were cisgender, since the focus of the current study lay elsewhere.

Upon completion, participants received £2.50. This payment was based on the £7.50 per hour participation fee recommended by Prolific Academic. The time to conduct the study was estimated at 20 minutes. By following recommended payment rates, we could expect that dropouts would not depend on the amount of payment. To determine the sample size for Phase 1, we conducted an a priori power analysis ($\alpha = 0.05$, d = 0.5, $\beta = 0.80$, one-tailed independent sample t-test), which resulted in 102 narrations in total, or 51 per type of violence to reach a power of approximately.80.

Phase 1 was completed by 113 narrators. However, 42 were excluded because they did not follow the instructions regarding writing the narratives of violence, either because their narratives were shorter than the minimal required length, or that they lacked description of violence as defined by WHO's definition of IPV. The sex ratio of participants that generated excluded statements were similar to the sex ratio of participants that generated included statements. This evaluation was made by two authors of this paper, disagreements were discussed and

resolved. Thus, it would not have been meaningful to keep these statements as they either lacked information, or had insufficient information to evaluate violence from.

Each participant wrote one narrative about physical violence and one about psychological violence. The final data consisted of 68 narrations about physical violence and 68 narrations about psychological violence. These narratives were collected from a total of 71 narrators, as some of narrators only produced one narrative and others produced two (aged 20–72 years, $M = 34.55$ years, $SD = 11.92$ years, 49 men, 22 women).

*Material and procedure.* The data were collected using an online questionnaire about IPV. Participants were recruited from the US. Narrators were asked to describe an event they had experienced, occurring in a close relationship, where they were victims of psychological or physical violence. They were asked to describe the event as they would to a close friend, as clearly and with as much detail as possible. In order to get sufficient information to evaluate the seriousness of the statement, the participants were required to write a minimum of 50 words. If the participants had not experienced any psychological or physical violence, they were instructed to describe a situation that was as close to this violence as possible. This was done to facilitate the generation of narratives with a low severity of violence. The generated narrations included descriptions of the violent event, which typically included description of the physical, or the psychological, violence they suffered. Furthermore, the narrations typically included the event that in their view caused the violent act, as well as the violent act itself.

There was no reference to time, so the participants could describe an event from any time of their lives, and were not asked for time of the event. We provided no specific definition of the relevant concepts 'physical violence,' 'psychological violence' or 'seriousness of violence'. This choice was made because our main focus was to study *how the severity of these concepts was communicated, rather than how the concepts are defined.* This allows for an empirically grounded usage of these concepts, where we can monitor the difference in severity ratings of these concepts for people experiencing the events related to the concepts and people receiving text descriptions of the events. To be clear, we understand that the concepts used can be interpreted differently depending on individual differences and backgrounds of the tested population. For example, the concept 'seriousness of violence' could be interpreted differently depending on how participants emphasize the effects of violence, related to emotional suffering, physical suffering, legal consequences, social consequences, etc., and on long or short timescales. Our purpose was not to provide an exact definition of these concepts, but to study what ratings the concepts evoked in the participants, given that the participants in the phase 1 and 2 were generated from the same population.

The order of descriptions of violence were counterbalanced between participants, so half of the participants were first asked about physical violence followed by the question of psychological violence, and the other half were asked in the opposite order. After the completion of each narration, they were asked to rate the seriousness of the event on a Likert scale, ranging from 0 = *not serious at all*, to 10 = *very serious*. Finally, demographic and relationship information was collected, such as duration of the relationship. The participants were debriefed with the information that they could contact a health professional, given that their response had evoked negative emotional reactions.

*Results and discussion.* We report the results from the 68 physical violence and the 68 psychological violence narrations that were retained after the exclusion described above. A dependent samples t-test showed that there was no difference in the severity ratings between the narrations of psychological violence ($M = 6.19$, $SD = 2.33$) and physical violence ($M = 6.85$, $SD = 3.01$, $t(67) = 0.20$, $p = .841$). As we were interested in potential Gender PD, we also checked whether there was a gender difference in the ratings. This main effect did not reach conventional levels of significance in a repeated measures ANOVA ($p = .051$). The small

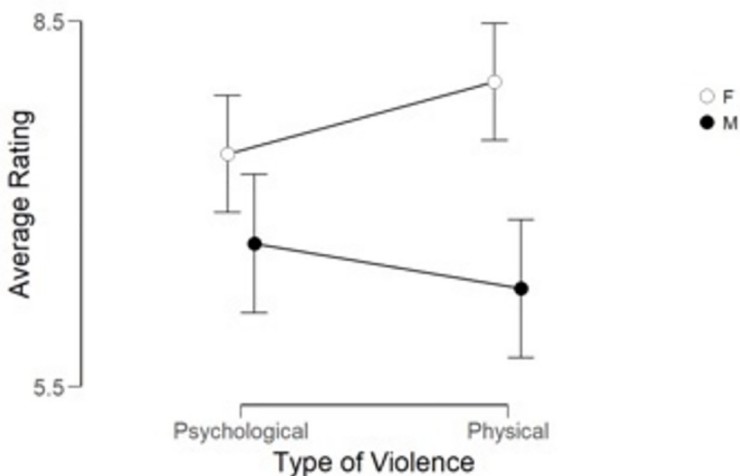

**Fig 2. Narrator ratings of severity of violence divided by type of violence and gender in Phase 1.** *Note*. F (hollow circle) stands for female and M (filled circle) for male gender, psychological violence is on the left side and physical on the right side. Bars show 95% confidence interval.

sample size, and the uneven number of men and women, may have contributed to the lack of significant differences. The mean ratings by narrator and gender are shown in Fig 2, and the accompanying ANOVA table in Table 1.

**Phase 2: Raters rate narrations from Phase 1.** *Participants*. For Phase 2, 340 participants were recruited (170 Women, 18–73 years, $M = 34.95$ years, $SD = 12.18$ years) from the same panel of participants as Phase 1. The same inclusion criteria from Phase 1 were used. Participants in Phase 1 were excluded from Phase 2.

*Material and procedure*. The second questionnaire consisted of making seriousness ratings of narrations from Phase 1. The questionnaire included the same information as in Phase 1, except no narratives were collected. The written narrations collected in Phase 1 were distributed into 17 groups of eight narrations: four of physical violence and four of psychological violence. The four physical-violence narrations were from different individuals than the four psychological narrations. For example, those who rated the psychological narrations from group 1 rated the physical narrations from group 2. To control for gender, ten men and ten women rated each narration. The order of the presentations of the physical and the psychological violence severity ratings were counterbalanced across participants, so the two types of

**Table 1. ANOVA tables for narrator ratings by type of violence, Phase 1.**

**Within Subjects Effects**

| | Sum of Squares | df | Mean Square | F | p | $\eta^2$ |
|---|---|---|---|---|---|---|
| Type of Violence | 0.365 | 1 | 0.365 | 0.128 | 0.722 | 0.002 |
| Type of Violence ✳ Gender | 6.865 | 1 | 6.865 | 2.410 | 0.125 | 0.035 |
| Residual | 188.018 | 66 | 2.849 | | | |

**Between Subjects Effects**

| | Sum of Squares | df | Mean Square | F | p | $\eta^2$ |
|---|---|---|---|---|---|---|
| Gender | 43.97 | 1 | 43.97 | 3.958 | 0.051 | 0.057 |
| Residual | 733.15 | 66 | 11.11 | | | |

*Note*. Gender refers to the gender of the narrator, type of violence is either physical or psychological. Type III Sum of Squares

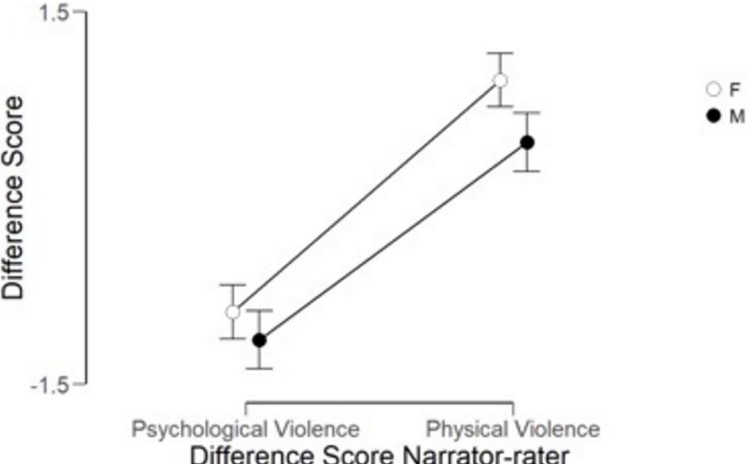

**Fig 3. Difference score between severity ratings of violence for narrator and rater in Phase 2.** F (hollow circle) stands for female and M (filled circle) for male gender, psychological violence is on the left side and physical on the right side. Bars show 95% confidence intervals.

ratings evenly distributed through the survey. Each rater was asked to read eight narrations and rate the seriousness of each event on a Likert scale (0 = *not serious at all*; 10 = *very serious*). In all other aspects the procedure was the same as for Phase 1.

*Results and discussion.* **Calibration PD.** To test for Calibration PD (H1), we first created difference scores by subtracting the narrator's seriousness rating from each rater's rating. A negative score indicates that the raters rated the event as less serious than the narrator, whereas a positive score indicates that the event was rated as more serious by the raters. This resulted in four difference scores for psychological violence, and four difference scores for physical violence, for each rater. These were averaged into two single scores–one for physical violence and one for psychological violence. The scores were analyzed in a repeated measures ANOVA, with gender of rater as a between-subjects variable. Means are shown in Fig 3, and the accompanying ANOVA table in Table 2. Raters overrated the narrations of physical violence (M = 0.70, SD = 1.74) and underrated the narrations of psychological violence compared to the narrators' ratings (M = -1.03, SD = 1.94, F(1,338) = 229,2, p < .001). Male and female raters

**Table 2. ANOVA tables for difference score between narrator and rater by type of violence, Phase 2.**

**Within Subject Effects**

| | Sum of Squares | df | Mean Square | F | p | $\eta^2$ |
|---|---|---|---|---|---|---|
| Difference Score Narrator-rater | 508.015 | 1 | 508.015 | 229.197 | < .001 | .403 |
| **Difference Score Narrator-rater** | | | | | | |
| ✳ **Gender** | 3.214 | 1 | 3.214 | 1.450 | .0229 | 0.003 |
| **Residual** | 749.177 | 338 | 2.217 | | | |

**Between Subjects Effects**

| | Sum of Squares | df | Mean Square | F | p | $\eta^2$ |
|---|---|---|---|---|---|---|
| Gender | 22.16 | 1 | 22.158 | 4.886 | 0.028 | 0.014 |
| Residual | 1532.76 | 338 | 4.535 | | | |

*Note*. Gender refers to the gender of the narrator, Difference Score refers to the rated severity of violence of the raters minus the severity score of the narrators. Type III Sum of Squares

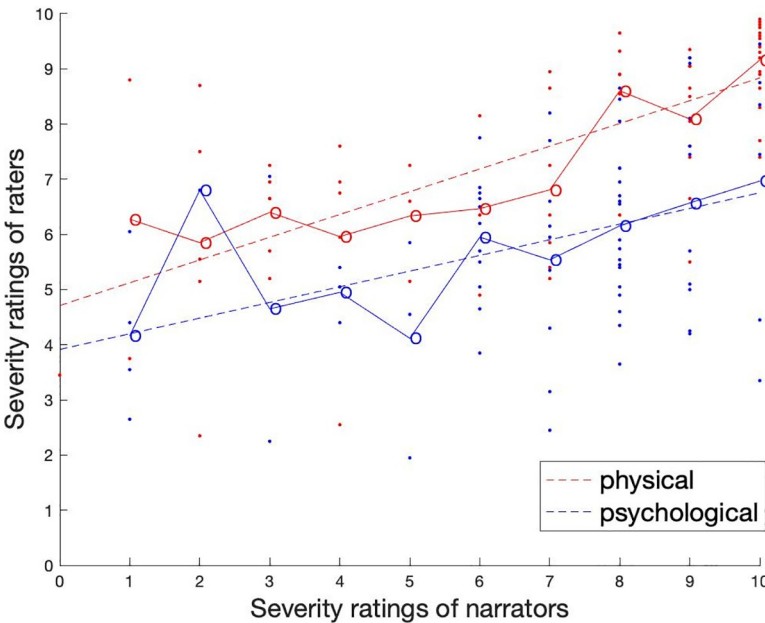

**Fig 4. The severity of violence as evaluated by the raters as a function of the evaluation of the narrators for all participants (men and women) in Phase 2.** The data are divided into physical (red) and psychological (blue) violence. Each dot (N = 340) represents one narration. The straight dotted lines are fitted linear curves. The circles with solid lines represent averaged severity ratings.

differed in their average ratings. There was no interaction between type of violence and gender of rater.

*Accuracy PD.* To analyze the Accuracy PD, we used the mean score of the raters for each narration (as opposed to the difference score). The Accuracy PD (H2) was tested by correlating the narrator's rating with the mean of all raters' ratings of the same event (Fig 4). For narrations about psychological violence, the overall Pearson's correlation was $r$ (n = 67) = .37, $p$ = .002. For narrations about physical violence, the overall Pearson's correlation was $r$ (n = 67) = .67, $p < .001$. A Fisher's z-test showed that the Pearson's correlations differed significantly, $z = -2.41$, $p = .016$, 95%-CI [-0.55, -0.05].

The interrater reliability was also measured by ICC [38]. We included this measure because, in contrast to Pearson's r, it accounts for the differences in ratings for individual correlation between raters. For raters, the ICC for narrations about psychological violence was poor, for example, 0.383 with a 95% confidence interval from 0.30 to 0.48 ($F(67,1292) = 13.4$, $p < .001$). The interrater reliability was fair for narrations about physical violence where the ICC was 0.465, with a 95% confidence interval from 0.38 to 0.56 ($F(67,1292) = 18.4$, $p < .001$).

**Phase 3: Ratings of narrations with manipulated gender in the narratives.** *Participants.* For Phase 3 of the study, 340 new raters (170 women, 18–75 years, $M = 35.40$ years, $SD = 7.83$) were recruited. They were recruited from the same panel as participants from Phase 1 and 2, however, previous participants were excluded from this phase.

*Material and procedure.* The same questionnaire and procedure used were the same as in Phase 2, but with the following exception: the gender of the narrators was swapped by exchanging the pronouns (i.e., 'he' was replaced with 'she,' and 'she' replaced with 'he'). Narrations where it was impossible to change gender were excluded from the analysis, which included narrations involving pregnancy (because of the impossibility of pregnant men) or

where pronouns were missing (because gender could not be identified in these narratives). This resulted in a reduction of usable narrations to 61 pairs, from the original 68 pairs.

As in Phase 2, the narrations were divided into 17 groups. However, the number of narrations per group and the type of violence were unequal, ranging from five narrations to only two narrations per group. In each group, the physical narrations and psychological narrations were from different narrators. Each narration was rated by 10 men and 10 women, with the exception of two that were rated by 10 women and 9 men.

*Results*. The Calibration and Accuracy PDs were hypothesized to be found regardless of gender, and therefore we tested if these perceptual differences occurred before investigating the Gender PD.

*Calibration PD*. Difference scores were calculated in the same manner as in Phase 2. Note that the variation in number of ratings that each rater completed may introduce additional noise in the results. Means can be found in Fig 5, and the ANOVA tables in Table 3. Consistent with the findings from Phase 2, raters overrated the narrations about physical violence ($M = 1.45$, $SD = 2.10$), showing a Calibration PD in the predicted direction. Furthermore, the difference scores were higher for physical than psychological violence ($M = 0.20$, $SD = 2.15$). However, the difference scores for psychological violence narrations were not significantly different from 0 ($p = .093$), suggesting that there was no Calibration PD for psychological violence. However, another possible interpretation that cannot be ruled out by the data is that reversing the gender made raters see the events as more severe. The gender reversal manipulation may simply have moved all of the difference scores up towards the positive side. As in Phase 2, male and female raters differed, in that female raters rated psychological violence as more serious than men. Perhaps more important is the interaction between type of violence and gender of rater, $F(1,356) = 14.26$, $p < .001$, $\eta^2 = .033$. As can be seen in Table 3, the difference in the ratings between the two types of violence is larger for the female raters than the male raters.

***Accuracy PD***. The predicted Accuracy PD was found, which is consistent with Phase 2. The correlation between the narrator ratings and the average ratings was larger for physical violence, $r$ (n = 61) = .529, $p < .001$ than for psychological violence, $r$ (n = 61) = .328, $p < .01$.

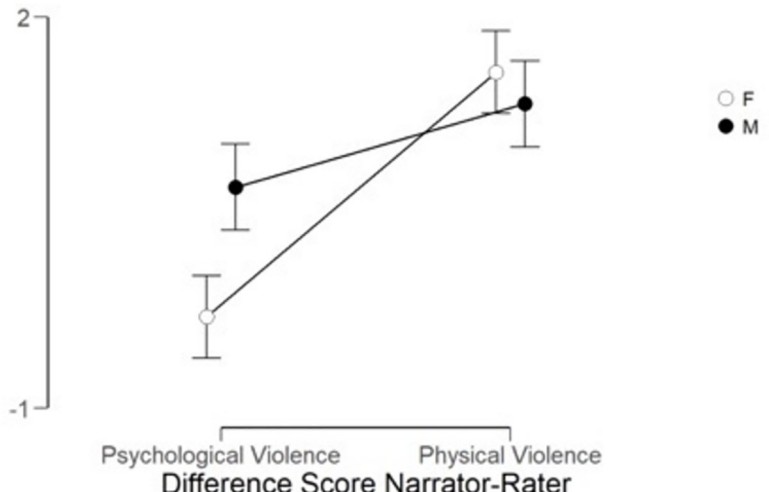

**Fig 5. Difference score between severity ratings of violence for narrator and rater in Phase 3 using gender-reversed narrator stories.** F (hollow circle) stands for female and M (filled circle) for male gender, psychological violence is on the left side and physical on the right side. Bars show 95% confidence interval.

**Table 3. ANOVA tables for difference score between narrator and rater by type of violence, gender reversed narratives, Phase 3.**

**Within Subject Effects**

| | Sum of Squares | df | Mean Square | F | P | $\eta^2$ |
|---|---|---|---|---|---|---|
| Difference Score Narrator-rater | 282.93 | 1 | 282.931 | 59.04 | < .001 | 0.13 |
| Difference Score Narrator-rater | | | | | | |
| ✻ Gender | 68.32 | 1 | 68.320 | 14.26 | < .001 | 0.03 |
| Residual | 1706.02 | 356 | 4.792 | | | |
| **Between Subjects Effects** | | | | | | |
| | Sum of Squares | df | Mean Square | F | p | $\eta^2$ |
| Gender | 25.20 | 1 | 25.203 | 6.288 | 0.013 | 0.017 |
| Residual | 1426.94 | 356 | 4.008 | | | |

*Note.* Gender refers to the gender of the narrator, Difference Score refers to the rated severity of violence of the raters minus the severity score of the narrators. Type III Sum of Squares

***Gender PD***. For analyzing Gender PD (H3), we averaged the ratings for each narrative across raters of the same gender, across both Phase 2 and Phase 3. The means for narratives that could not be used for the gender-reversal in Phase 3 were also removed from the ratings for Phase 2, so that the ratings were based on the same narratives. The average ratings were then analyzed in an omnibus ANOVA with the following independent variables: type of violence, gender manipulation, narrator's gender, and rater's gender. Because this involves multiple comparisons, the alpha level was set to 0.01. The full ANOVA table can be found in Table 4.

There were three main effects. First, raters rated types of violence differently, with physical violence rated as more serious than psychological violence (M = 7.90, SD = 1.59; M = 6.48, SD = 1.89). Second, the gender manipulated narrations were rated as more serious than the original narrations (M = 7.69, SD = 1.52; M = 6.69, SD = 2.08). Third, the gender of the narrators influenced how the raters perceived the seriousness. The original female narrations were rated as more serious than original male narrations (M = 7.56, SD = 1.83; M = 7.01, SD = 1.88). However, the rater's gender had no effect of (*p* = .968). There was also an interaction between gender manipulation and narrator, which we further analyzed separately for psychological and physical violence using gender manipulation and perceived gender as the independent variables.

**Table 4. ANOVA table for rating of seriousness by type of manipulation (original, gender reversed), perceived gender of the narrator, and rater gender.**

**Within Subject Effects**

| Cases | Sum of Squares | df | Mean Square | F | P | $\eta^2$ |
|---|---|---|---|---|---|---|
| TypeOfViolence | 189.399 | 1 | 189.399 | 73.009 | < .001 | 0.118 |
| **Manipulation** | 70.982 | 1 | 70.982 | 27.362 | < .001 | 0.044 |
| PerceivedGender | 29.394 | 1 | 29.394 | 11.331 | < .001 | 0.018 |
| **RaterGender** | 0.004 | 1 | 0.004 | 0.002 | 0.968 | 0.000 |
| TypeOfViolence ✻ Manipulation | 12.166 | 1 | 12.166 | 4.690 | 0.031 | 0.008 |
| TypeOfViolence ✻ PerceivedGender | 1.284 | 1 | 1.284 | 0.495 | 0.482 | 0.001 |
| TypeOfViolence ✻ RaterGender | 13.050 | 1 | 13.050 | 5.031 | 0.025 | 0.008 |
| Manipulation ✻ PerceivedGender | 32.325 | 1 | 32.325 | 12.461 | < .001 | 0.020 |
| Manipulation ✻ RaterGender | 13.531 | 1 | 13.531 | 5.216 | 0.023 | 0.008 |
| PerceivedGender ✻ RaterGender | 0.571 | 1 | 0.571 | 0.220 | 0.639 | 0.000 |

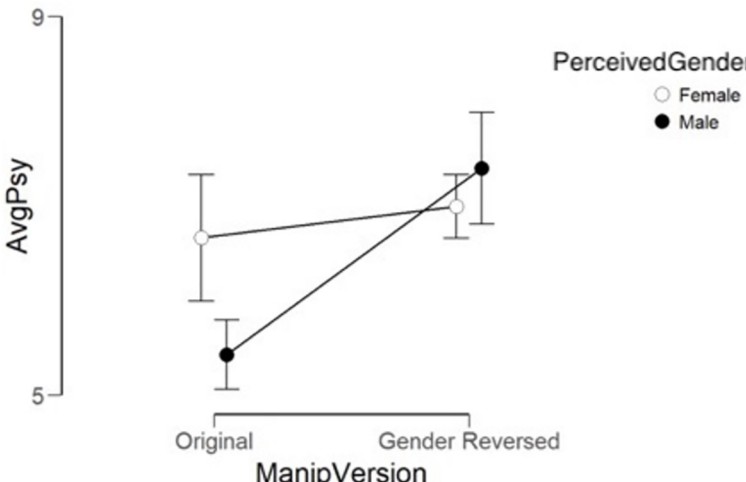

**Fig 6. Comparison of ratings of original narratives and gender-reversed narratives by perceived gender and psychological violence.** Hollow circles stand for female and filled circles for male gender, original ratings are on the left side and gender reversed on the right. Bars show 95% confidence interval.

*Psychological violence*. As is indicated in Fig 6, and Table 5, ratings in the gender manipulated narratives were higher overall than in the original narratives. There was also an interaction between perceived gender and manipulation. In the original version, the female narratives were rated as more severe than the male narratives. In the gender-reversed stories, the difference in severity-ratings were higher and more similar between the perceived genders. An analysis of simple main effects showed that whereas there was no difference in severity ratings for the female narratives in the two versions, a significant difference was found between the more severally rated male narratives in the gender reversed condition, and the original version.

*Physical violence*. Fig 7 and Table 5 also show a main effect of version in the same direction as for the ratings of Psychological Violence. In addition, there was a main effect of perceived sex of the narrator where ratings for female narratives were overall rated as more severe than those of male narratives. However, there was no interaction between version and perceived gender, thus no simple effects analysis was performed.

Finally, to further understand the possible effects of the gender manipulation, we used independent samples t-tests to compare the original ratings with the gender manipulated ratings. When the gender was changed from female to male, there were no significant differences in the ratings for either type of violence. When gender was changed from male to female, however, raters rated the gender manipulated narrations as more severe for both psychological violence, $t(82) = 6.26$, $p < .001$, and for physical violence, $t(82) = 4.61$, $p < .001$.

**Table 5. ANOVA comparing ratings of narratives of psychological violence for the original version, and the gender reversed version (Phase 2 and 3).**

**Within Subject Effects**

| Cases | Sum of Squares | df | Mean Square | F | P | η² |
|---|---|---|---|---|---|---|
| Manipulation | 70.961 | 1 | 70.961 | 23.740 | < .001 | 0.085 |
| PerceivedGender | 9.195 | 1 | 9.195 | 3.076 | 0.081 | 0.011 |
| Manipulation ✱ PerceivedGender | 36.021 | 1 | 36.021 | 12.051 | < .001 | 0.043 |
| Residual | 717.369 | 240 | 2.989 | | | |

**Note.** Manipulation is either original texts or manipulated texts and PerceivedGender is the gender in the manipulated narrations. Type III Sum of Squares

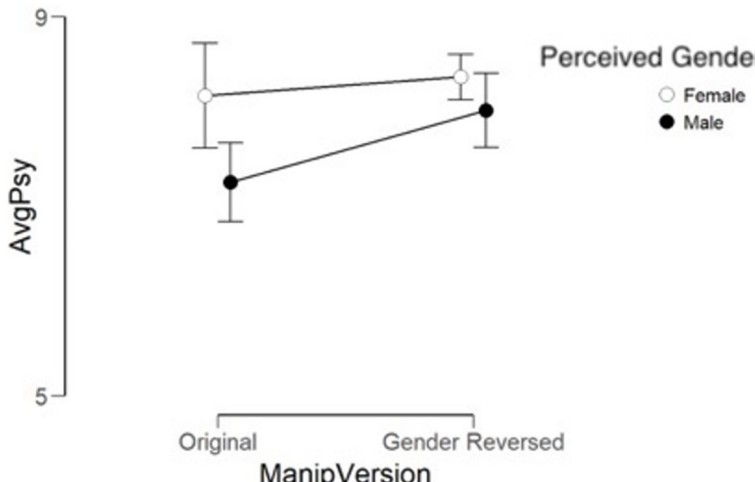

**Fig 7. Comparison of ratings of original narratives and gender-reversed narratives by perceived gender and physical violence.** Hollow circles stand for female and filled circles for male gender, original ratings is on the left side and gender reversed on the right side. Bars show 95% confidence interval.

*Effect size of the perceptual differences*. Our data also allow estimation of effect size and/or simulation of the possible implication of these perceptual differences. By using Signal Detection Theory (SDT), it is possible to estimate how likely it is that a communicated violent act is sufficiently severe to pass a threshold for punishment (for example in a court trial, or as being offensive in social settings) depending on whether it is perceived as being conducted by a man or by a woman. For example, a narrative describing a man as the victim of physical violence (i.e., written by a man in Phase 3) has a likelihood of 36% chance of being evaluated to have a severity above a certain criterion. However, when the same stories are changed so that the woman is believed to be a victim, then this likelihood almost doubles to 64% (i.e., an increase of 30% where the criterion is here placed symmetrically between the values of the two conditions, with an average value of 8.0). This suggests that when a man is exposed to the same physical violence as a woman, he is considerably less likely to be seen as a victim. This effect cannot fully be explained by the fact that men possess greater physical strength and are therefore (independent of the evidence) less likely to be victimized by a woman, because this explanation does not account for the fact that the same effect is found for psychological violence. For example, narratives written by men have a 43% chance of being evaluated to have severity above a certain criterion, whereas the same stories have a 57% chance of passing the same criterion when they are perceived to be narrated by a woman, i.e., an increase of 15%. A similar effect can be found for the Calibration PD, where psychological violence has a 60% probability of being above the criterion for the person experiencing the event, and 40% for the person reading about it.

**Discussion.** Phase 3 uncovered some surprising effects. The main purpose was to investigate possible Gender PDs, in which the same narrations would be judged differently if the perceived gender was manipulated. This was supported in cases where the gender of the narrator was manipulated from male to female.

An overall Calibration PD was found, where the seriousness ratings in gender-manipulated narratives were rated as more serious than for original narratives. This perceptual difference was primarily found for physical violence, when narratives of men were manipulated to be

perceived as narratives describing women. However, for psychological violence no Calibration PD was found.

## General discussion

The aim of this research was to look for potential perceptual differences in the communication of IPV in heterosexual romantic relationships. We expected to find Calibration PD (H1), Accuracy PD (H2), and Gender PD (H3). It is important to note that we d

o not necessarily claim that either of the ratings are 'correct,' as measurement errors may occur both on the part of the narrator of the event, and the rater. At the same time, the narrator has more knowledge of the original event, which is why we assume that their rating is more 'accurate.'

A Calibration PD (H1) was found, where raters rated narrations about physical violence as more serious than narrators did. Additionally, the narrators' seriousness ratings of psychological violence were higher than the independent raters' ratings, also supporting the Calibration PD in the Phase 2 data. There may be several explanations for this result. One possibility is that psychological violence is harder to communicate because it is based on secondary reinforcement, i.e., a learned stimuli previously associated with a primary reinforcer or a stimulus that satisfies a basic survival instinct, such as physical violence. This secondary reinforcement may make psychological violence harder to detect for the receiver of the information. In the Phase 3 data, there was no Calibration PD for psychological violence, although there was a difference in calibration between physical and psychological violence. A possible reason for this is that the Phase 3 data also show that violence towards women is seen as more severe, which may have biased the results, as more narrations were manipulated to be 'female' in this phase.

An Accuracy PD (H2) was confirmed, with lower correlations between narrator's rating and the mean of all raters' ratings for narrations about psychological violence, compared to physical violence. This finding is further supported by the higher interrater reliability for ratings on narrations about physical, as compared with psychological violence. This supports our hypothesis that psychological violence is harder to communicate than physical violence. A possible reason for the lower accuracy in the ratings of psychological violence could be that psychological violence relies on a secondary reinforcement that is unique to each person, or an ambiguity in the conceptualization of this type of violence.

Investigating the Gender PD (H3), our results indicate that narrations written by females about physical violence were rated as more serious by both male and female raters. Thus, the hypothesized Gender PD was confirmed: gender stereotypes of the narrator influence perception. In order to establish whether or not this was because of contextual/language differences, or because of gender, we compared the original narratives written by females with the narrations manipulated into male, and found no significant difference, indicating no Gender PD for narrations originally written by women. However, when we made the same comparison for male narrators, a significant difference was found, where the manipulated narrations (i.e., originally written by a male and manipulated into a female) were rated as more serious than the original. This may indicate that for narrations written by males, raters take the language and context into account. The narrations originally written by a male included a context or used words and descriptions which were rated as more serious when perceived as being written by a female. In this context it should be noticed that previous studies show that females are generally better at expressing themselves than males [39]. This allows them to communicate the context and course of action more clearly. Furthermore, male narrators try to minimize their victimization [34], resulting in them using less accurate descriptions of experienced violence.

Another interesting finding was that serious ratings of females' narrations about physical violence and males' narrations about psychological violence did not differ significantly when

rated by males or females. This consensus between the genders consisted of females' narrations about physical violence being rated as the most serious, and males' narrations about psychological violence being rated as the least serious. The finding that men experiencing psychological violence is perceived as the least serious is especially alarming, since Prospéro [40] found that men suffer severe consequences from this type of IPV.

## Practical implications

This is the first study using our experimental procedure, and the results must be corroborated. However, given that our findings can be replicated, we believe that our results and data provide useful insights into the communication and perception of violence in romantic relationships, and can therefore be a helpful tool when statements made by plaintiffs, witnesses, or other parties are to be evaluated. Below we summarize some concrete implications and practical guidelines for readers. In some contexts, evaluators need increased sensitivity, i.e., to say that that a situation is violent although the severity of the evaluated violence may fall below their accepted threshold for saying this; in other contexts, evaluators may need reduced sensitivity, i.e., to say that no violence occurs, although violence may exceed the severity evaluation threshold:

General guidelines

- *Evaluation of severity of violence depends on several factors*. These factors include: type of violence, whether violence is experienced or communicated, and the gender of the people involved. Evaluations of severity must account for each of these factors and their reliability.

Guidelines regarding rating other people's texts of psychological violence

- *Increase sensitivity during evaluation of psychological violence*. Results from perceptual differences in calibration of psychological violence indicate that people tend to underestimate the severity of psychological violence in close relationships. Therefore, be extra careful and respectful while reading texts that relate to other people's communication of psychological violence.

- *Be aware about uncertainties during evaluation of psychological violence*. Our data suggest that it is a difficult task to evaluate the severity of psychological violence. Therefore, be humble about your evaluation, as you are likely to evaluate it either as more, or less, severe than actually perceived by the victim.

Guidelines regarding ratings of other peoples' texts of physical violence

- *Decrease sensitivity during evaluation of physical violence*. Our data indicates that expressions of physical violence may sound worse than how they are actually perceived by the person experiencing the event. Therefore, be careful not to overestimate the severity of physical violence communicating by other people.

Guidelines regarding gender

- *Increase sensitivity for male victims*. Our data show that narratives with male victims tend to be evaluated as less severe than the same narratives with female victims.

- *Increase sensitivity of male, compared to female, raters*. If you are male rater, or are given evaluations of violence from a male rater, then consider increasing your sensitivity, compared to a female, or one given evaluations from a female rater. Generally speaking, our data indicate that males tend to give lower seriousness ratings than females.

Guidelines to victims of violence

- People to whom you communicate psychological violence may be poor at understanding the severity of the violence. Therefore try to be very clear regarding whether the violence is severe, or not. Otherwise, the severity of the event may not get across to the person to whom you are communicating it, and it may also be viewed as less severe. If you are male, then the evaluator may view the violence as being less severe than it actually is, so violence done to you may not be taken at face value.

The guidelines listed here may have practical use in everyday life and professional settings. For example, it may help social workers to assess inter-family physical violence and psychological violence, such as blaming the other parent or withdrawing contact between family members. Similarly, it may be used by police and judges to evaluate the severity of domestic violence, with application to family law, and custody cases, but also to criminal cases.

Because the Calibration PD is a systematic error, it should at least theoretically be possible to correct by simply subtracting (in physical violence) or adding (in psychological violence Phase 2 data) a factor of seriousness ratings when independent raters read descriptions of violence. However, to what extent this is practically possible is an open question that must be addressed with additional data. It also provides the practical guideline that we should have greater trust in the original ratings of the narrator of psychological violence, and more humility in making interpretations of severity ratings made by a person not exposed to this type of violence.

This study sheds light on the potential extent of gender differences, and can raise awareness of these perceptual differences among the general public. Studies suggest that the most common reason women do not seek help is minimization of the incident, as well as feelings of isolation, shame, and fear of being judged by others [41, 42]. Knowledge about the perceptual differences in communication of IPV may improve the fairness of laws and policies, as well as institutional responses to different types of violence. For example, training individuals to better identify IPV can lead to more appropriate reactions and targeted treatment for both victims and perpetrators [14].

## Limitations

The current study has several limitations. The ratings of severity of violence were conducted with only one item, where it is possible that the reliability of this measure would have been improved with additional items, using a validated scale of severity of violence. The interpretation of the severity scale may also influence the results. A 10 may have been interpreted differently by the participants (e.g., 'The most severe that I have ever experienced with a partner' or 'The most severe violence that I could imagine anyone might experience'). Although participants were instructed to produce a minimum number of words (i.e., 50), the texts generated were also rather short. It is possible that longer texts may have decreased some of the perceptual differences. Furthermore, the current study is limited to written text data; to what extent the current study generalizes to speech data is not known. Future studies may look into the difference between text and verbal data using the current methodology.

The current study is limited to studying the distinction between physical and psychological violence. Future studies may look into finer distinctions, for example, to what extent sexual violence differs from physical violence. Sexual violence may give rise to other types of psychological trauma that, in interaction with the gender of victims, may influence the severity evaluation of the violence. Both physical and psychological violence can be further divided into various subtypes not investigated in the current study. Furthermore, the participants were not informed, or educated about, various types of violence. For example, they were not introduced

to the distinction between situational couple violence, intimate terrorism, and violent resistance.

The current study uses a quantitative method where participants use rating scales to evaluate the severity of violence. Complementary information may be acquired by adding a qualitative component. For example, participants could be interviewed to obtain details regarding their experiences of violence, which for example may facilitate understanding of the different findings related to psychological and physical violence.

The different distributions of the original ratings may also play a role. There are many more physical violence narrations rated at the upper boundary (i.e., 10) than psychological narrations, for both men and women, suggesting a ceiling effect. There were twice as many original narrations with male narrators compared to female narrators. This does not necessarily bias the results, but it means that we have a much higher sampling for the male narratives, that is, the mean estimates are more robust. The clusters vary on gender composition and mean severity, which in turn introduces error into our measures. However, the difference scores were normally distributed. The grouping of narratives into smaller clusters was done as an attempt to randomize the narratives across raters, but does not provide a complete randomization.

Another limitation is the selection of participants. The current study uses a general population recruited from Prolific Academic. It would be of interest to study if the findings generalize to professionals that work in the criminal justice or legal systems. This is a potential focus of future studies.

Furthermore, the current study does not investigate the race of the persons experiencing the violent act, and how this interacts with the race of the evaluators. For example, the seriousness ratings may very well depend on whether the victim is Black or white, and whether the evaluator is of the same or a different race. However, the narrations produced in the current study typically did not include information about the race of the victim or the offender, nor did we collect data on the race of the participants of the study, so it was not possible to analyze how race interacted with the ratings. It should be noted that the proposed method does not measure an 'objective' severity of violence. Rather, it compares the subjective severity of the victim's experience of the event with the observed severity of the person that this event is verbally communicated to.

## Conclusion

The main findings of our study were that Calibration and Accuracy PDs were found in the communication of physical and psychological violence. Furthermore, a Gender PD was found, indicating that narratives written by females were on average rated as more serious by both male and female raters. This was also found when the male narrator was believed to be female (original narrator male). Moreover, we found that male raters rated according to the stereotypical masculine gender norm, which led them to rate other males' narrations as the least serious. However, they rated narrations that they believed were written by a male as more serious (original narrator female). With respect to the practical application of our findings, we believe that the gained insights can be used to raise awareness of different types of PDs in relation to different types of IPV.

## Author Contributions

**Data curation:** Hannah Lettmann, Anna Alexandersson, Elena Schwörer.

**Formal analysis:** Hannah Lettmann, Anna Alexandersson, Elena Schwörer, Åse Innes-Ker.

**Investigation:** Sverker Sikström.

**Methodology:** Sverker Sikström.

**Project administration:** Sverker Sikström.

**Supervision:** Sverker Sikström.

**Validation:** Sverker Sikström.

**Writing – original draft:** Sverker Sikström, Mats Dahl, Hannah Lettmann, Anna Alexandersson, Elena Schwörer, Lotta Stille.

**Writing – review & editing:** Sverker Sikström, Mats Dahl, Lotta Stille, Oscar Kjell, Åse Innes-Ker, Leonard Ngaosuvan.

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
