## [Decision Letter · Decision Letter 0]

3 Dec 2020

PONE-D-20-32794

What You Say and What I Hear – Investigating Differences in the Perception of the
Severity of Psychological and Physical Violence in Intimate Partner
Relationships

PLOS ONE

Dear Dr. Sikström,

Thank you for submitting your manuscript to PLOS ONE. After careful consideration, we
feel that it has merit but does not fully meet PLOS ONE’s publication criteria as it
currently stands. Therefore, we invite you to submit a revised version of the
manuscript that addresses the points raised during the review process.

Thank you for submitting your manuscript to PLOS ONE. Your paper has been reviewed by
two independent reviewers and was found to have several issues of content and
format, including aspects related to the conceptualization of seriousness and
typographical errors. The topic is appealing to the readership and fills a gap in
research knowledge. After considering the reviewers' decisions, this Academic Editor
consider that major revisions are needed. Please address the reviewers'
recommendations and resubmit for further consideration. The manuscript cannot be
considered for publication in its current form. 

Please ensure that your manuscript meets PLOS ONE’s publication criteria

Please submit your revised manuscript by Jan 16 2021 11:59PM. If you will need more
time than this to complete your revisions, please reply to this message or contact
the journal office at plosone@plos.org. When
you're ready to submit your revision, log on to https://www.editorialmanager.com/pone/ and select the 'Submissions
Needing Revision' folder to locate your manuscript file.

If you would like to make changes to your financial disclosure, please include your
updated statement in your cover letter. Guidelines for resubmitting your figure
files are available below the reviewer comments at the end of this letter.

We look forward to receiving your revised manuscript.

Kind regards,

Abraham Salinas-Miranda, MD, PhD

Academic Editor

PLOS ONE

Journal Requirements:

2. Please provide additional details regarding participant consent. In the ethics
statement in the Methods and online submission information, please ensure that you
have specified what type you obtained (for instance, written or verbal, and if
verbal, how it was documented and witnessed).

If your study included minors, state whether you obtained consent from parents or
guardians.

If the need for consent was waived by the ethics committee, please include this
information.

Additional Editor Comments:

Dear authors: Thank you for submitting your manuscript to PLOS ONE. Your paper has
been reviewed by two independent reviewers and was found to have several issues of
content and format, including aspects related to the conceptualization of
seriousness and typographical errors. Please address the reviewers' recommendations
and resubmit for further consideration. The manuscript cannot be considered for
publication in its current form.

Reviewers' comments:

Reviewer's Responses to Questions

**Comments to the Author**

1. Is the manuscript technically sound, and do the data support the conclusions?

Reviewer #1: Partly

Reviewer #2: No

2. Has the statistical analysis been performed
appropriately and rigorously? 

Reviewer #1: I Don't Know

Reviewer #2: Yes

3. Have the authors made all data underlying the
findings in their manuscript fully available?

Reviewer #1: Yes

Reviewer #2: Yes

4. Is the manuscript presented in an intelligible
fashion and written in standard English?

Reviewer #1: No

Reviewer #2: Yes

5. Review Comments to the Author

Reviewer #1: Thank you for the opportunity to review the manuscript entitled, What
You Say and What I Hear – Investigating Differences in the Perception of the
Severity of Psychological and Physical Violence in Intimate Partner Relationships.
The stated purpose of the study was to assess potential perceptual differences in
the communication of IPV in heterosexual romantic relationships. The authors stated,
"This study focuses on measuring perceptual differences in the communication and
perception of violence…To our knowledge these perceptual differences have not been
explored…"

1. I applaud the authors for their pioneering work and found the study's aim to be
relevant and important.

2. The manuscript would benefit for extensive editing to address several grammatical
and spelling errors, beginning with the first sentence in the Introduction, which
appears to have a typo, i.e. "A correction evaluation..". The authors may have
intended to write "A correct evaluation". Elsewhere, under guidelines, we read, " If
you are male rater, or are given evaluations of violence from a male rather, then
consider increasing your sensitivity. I believe the authors meant to write "rater" .
There are numerous other such examples throughout the manuscript and I encourage the
authors to edit their work.

3. There are several sections that would benefit from being more concisely written,
which would improve the work's readability. For example, the authors wrote,
"According to our hypotheses, we expected to find"… This could have been reduced to
"We hypothesized…"

4. I wonder if the relevance of the study would have been enhanced by adding raters
who actually work in the criminal justice or legal systems…This is because they are
likely to have experience which would inform the severity of their ratings.

5. The following sentence needs to be clarified: "Gender PD exists if the seriousness
ratings of violence depend on group belongingness of either the experiencer or the
rater".

6. The authors stated, "…Thus, statements about psychological violence can be
interpreted in more different ways across situations and individuals compared to
physical violence. For instance, sarcasm and irony makes it more difficult for the
receiver to fully comprehend the intention of the communicated statement. This leads
to lower inter-rater reliability in the evaluation of seriousness of psychological
violence".

Please clarify the leap from comprehension of the intention of a communicated
statement to inter-rater reliability.

7. I had hoped that the distinction among Situational couple violence ,intimate

Terrorism, and violent resistance would be included in the analysis. For example,
were the raters informed of the distinctions among these concepts? Did the authors
classify each narrative in these categories in order to determine which categories
of violence were rated as higher in severity, either by the "experiencer" or by the
rater?

8. Was sexual violence was rated as physical violence in this study. Research
indicates that sexual violence adds layers of psychological trauma that raters, in
this case, might rate as more severe, especially based on the gender of the
victim/survivor.

9. The authors stated, "First, the difficulty of communicating psychological violence
may lead to a Calibration PD where psychological violence is perceived as less
serious when it is communicated"…

This is an example of where complementary articulation of a study's findings would be
enhanced by adding a qualitative component. For example, by selecting a subsample of
participants for interviews from which the authors would obtain more detail
regarding their experiences of psychological violence.

10. The authors stated,"… psychological violence, being an indirect reinforcer, is
more difficult to communicate than physical violence… It is argued that this occurs
because psychological violence works as an indirect reinforcer, meaning that
psychological violence requires a learning process. This does not apply to physical
violence because it is a direct reinforcer…"

There is a great deal of literature that indicates social learning is involved for
both perpetrators and victims of IPV-

11. I have no problem with the fact that non-heteronormative persons were excluded
from this study. I am just not sure if the reason provided is sufficient, i.e., "The
screening of heterosexuality was conducted because we were interested in changing
the gender in the collected texts of violence, and we wanted to make sure that this
occurred in close relationships consisting of a man and a woman". Trans women, for
example, have a deeply felt internalized sense of being a woman.

12. "Upon completion, they received £2.50 for their participation". Please explain
how the authors arrived at this sum as an incentive and provide a reference. Under
limitations, include a discussion of whether or not this small incentive resulted in
the loss of so much data that out of 136 enrolled, the final sample was 71.

13. The authors stated, "The final data comprised of 68 narrations about physical
violence and 68 narrations about psychological violence. These narratives were
collected from a total of 71 narrators, as some of narrators only produced one
narrative and others produced two ".

Were there specific instructions regarding how many narratives were needed or was
there a space limitation?

14. From an ethical perspective, what steps were taken to mitigate participants'
psychological distress related to recall or re-experiencing adverse events such as
physical or psychological violence? Any debriefing? Referral for psychological
support as needed?

15. "In Phase 3: If the participants had not experienced any psychological or
physical violence, they were instructed to describe a situation that was as close to
this violence as possible".

I am not certain that the experience of being slapped, pushed down some stairs,
beaten severely or some other direct act of violence correlates with "a situation
that was as close to violence as possible".

16. The authors stated, "…This included narrations involving pregnancy (because it
would not make sense to have a story with pregnant men) or when pronouns were
missing (because gender could not be identified in these narratives)…"

I am confused, here. I thought part of the analysis was done by gender. What happened
to narratives for which gender could not be identified? Were they discarded?

17. The authors stated, "An analysis of simple main effects showed that whereas there
were no difference in severity ratings for the female narratives in the two
versions, but a significant difference in how the male narratives were rated with
the stories in the gender reversed condition rated as more severe than those in the
original version. This suggests that when a man is exposed to the same physical
violence as a woman, he is considerably less likely to be seen as a victim. This
effect can not fully be explained by the fact that men possess greater physical
strength and are therefore (independent of the evidence) less likely to be
victimised by a woman, because this explanation does not account for the fact that
the same effect is found for psychological violence".

The question which comes to mind is, aren't men often considered psychologically
stronger than women also? Based on neuroimaging studies involving the role of the
amygdala in emotional regulation Please see:

Eippert F, Viet R, Weiskopf N, Birbaumer N, Anders S. Regulation of emotional
responses elicited by threat-related stimuli. Human Brain Mapping. 2007;28:409–423.
[PMC free article] [PubMed]

See also:

McRae, K. et al. The neural bases of distraction and reappraisal. J. Cogn. Neurosci.
22, 248–262 (2010).

18. The authors stated, "One possibility is that psychological violence is harder to
communicate because it is based on secondary reinforcement, which may lead to it
being harder to detect by the receiver of the information" I think if the authors
provide a clearer definition of secondary reinforcement as they are using it, it
would clarify their point. ..

19. Under Guidelines, regarding gender, the authors recommend increasing sensitivity
but do not indicate how this would be done. Also, please check grammar and typos in
this section as well as others.

20. I was concerned about the following wording: "If you are male rater, or are
given

evaluations of violence from a male rather (Typo, please edit), then consider
increasing your sensitivity, compared to if you are female, or given evaluations
from a female rater.

Again, just recommending that someone increases their sensitivity does not constitute
a fait accompli; it's more likely to evoke cognitive dissonance. Guidelines need to
be accompanied by "how to" instructions.

21. I would omit the entire guidelines from this manuscript, as the intent was not to
write a manual. Stick to the purpose of the study. Your topic and findings are
important enough.

22. "Guidelines to victims of violence- If you are communicating psychological
violence, try to be very clear of whether you find this violence severe, or not.
This message may not get across the person you are communicating it to, and it may
also be viewed milder than what you feel that it actually was. If you are male, try
to emphasize the severity of violence more, the violence made to you may not be
taken at face value.

There is so much involved in reporting violence as a victim and so many phases one
must go through (See Transtheoretical model of change and IPV), and once the
decision is made to report IPV, the reporting itself is traumatic; So, the latter
quote can appear as if the authors are engaging in "Victim blaming". I am sure they
are not, but rewording that paragraph is important.

Other items that need to be addressed:

• DOIs missing from references

• Lots of grammatical and spelling errors throughout.

• Sometimes, no spaces are left after periods

• Sometimes, extra spaces are left between paragraphs

• Keeping "person" and "voice" consistent in the writing would be helpful. The
authors shift from first person plural to third person- and from active to passive
voice throughout the manuscript.

Reviewer #2: The authors aimed to examine perceptual discrepancies in narrators’ and
receivers’ understanding of the seriousness of IPV. They demonstrate that there are
some differences, and notably gender differences, in attributions of seriousness.
This study is novel and the notion of perceptual differences has implications for
both prevention programs as well as intervention services.

My main concern is the conceptual definition of the main construct, “seriousness.” It
is unclear what this concept means. The authors seem to indicate that it is
synonymous with severity as they use the terms interchangeably throughout. However,
how did the participants interpret the question regarding how “serious” the incident
was? Was there a pilot test with a debriefing session for participants prior to this
study? There are a lot of interpretations of the construct of seriousness that could
pose problems for this study in terms of reliability and what the findings mean. Is
it likelihood of acute physical or psychological trauma? Is it long-term health
effects? Does it tap into the extent to which the behavior is illegal or the extent
to which it should be punished by the criminal justice system? The authors mention
Johnson’s typology, is seriousness a reflection of the extent to which participants
think that the incident narrative is reflective of a relationship marked with
violent coercion compared to deficits in managing anger? With regard to gender
differences, one could respond that violence against women incidents are more
“serious” because they are embedded in larger structural systems that make economic
independence less likely and thus women may be less likely to be able to leave
abusive relationships. In sum, information regarding the validity and reliability of
the single item indicator of seriousness is needed as it could be open to much
interpretation.

The authors report that the sample is from the United States. Perceptions of injury
and harm are not only tied to gender, but strongly tied to race. Notably, Black
women are consistently perceived to be less impacted by violence, trauma, and health
issues. For example, their pain is perceived to be less by health practitioners in
hospitals and doctor’s offices. To what extent might these results be influenced by
race? Did the vignettes use any names, real or fake, to communicate the gender of
the narrator and the assailant? Was the distribution of the race of the narrators
and perceivers comparable? It is likely if there were no names used that the
perceivers mentally envisioned a scenario with two people of the same race as
themselves. If the distributions are comparable, I think that the authors can
reasonably conclude that this does not pose an issue, but it should be mentioned in
the methods and discussed in the discussion section.

Minor and Miscellaneous

- The authors mainly refer to gender differences, but in the methods they include
discussion of sex ratios. The authors report on sexual orientation, but not gender
identity. Were all participants cisgendered?

- The authors discuss Johnson’s typology. Although they mention violent resistance,
included in his expanded typology is also mutual violent control. Although a minor
omission, it is relevant given the emphasis of this study is on gender parity.

6. PLOS authors have the option to publish the peer
review history of their article (what does this mean?). If published, this will
include your full peer review and any attached files.

If you choose “no”, your identity will remain anonymous but your review may still be
made public.

**Do you want your identity to be public for this peer review?** For
information about this choice, including consent withdrawal, please see our
Privacy Policy.

Reviewer #1: No

Reviewer #2: **Yes: **Rachael Powers

Say and What I Hear _REVIEW 11 15 20.docx
---

## [Author Response · Author response to Decision Letter 0]

22 Jan 2021

Reviewers' comments and our responses:

Reviewer #1: Thank you for the opportunity to review the manuscript entitled, What
You Say and What I Hear – Investigating Differences in the Perception of the
Severity of Psychological and Physical Violence in Intimate Partner Relationships.
The stated purpose of the study was to assess potential perceptual differences in
the communication of IPV in heterosexual romantic relationships. The authors stated,
"This study focuses on measuring perceptual differences in the communication and
perception of violence…To our knowledge these perceptual differences have not been
explored…"

1. I applaud the authors for their pioneering work and found the study's aim to be
relevant and important.

Our comment: Thank you :)

2. The manuscript would benefit for extensive editing to address several grammatical
and spelling errors, beginning with the first sentence in the Introduction, which
appears to have a typo, i.e. "A correction evaluation..". The authors may have
intended to write "A correct evaluation". Elsewhere, under guidelines, we read, " If
you are male rater, or are given evaluations of violence from a male rather, then
consider increasing your sensitivity. I believe the authors meant to write "rater" .
There are numerous other such examples throughout the manuscript and I encourage the
authors to edit their work.

Our comment: These and other types are now corrected.

3. There are several sections that would benefit from being more concisely written,
which would improve the work's readability. For example, the authors wrote,
"According to our hypotheses, we expected to find"… This could have been reduced to
"We hypothesized…"

Our comment: The manuscript has been corrected accordingly. 

4. I wonder if the relevance of the study would have been enhanced by adding raters
who actually work in the criminal justice or legal systems…This is because they are
likely to have experience which would inform the severity of their ratings.

Our comment: We agree that this would be interesting, and have added this paragraph
in the limitation section: “Another limitation is the selection of participants. The
current study uses a general population recruited from Prolific Academic. It would
be of interest to study if the findings generalises to professionals that work in
the criminal justice or legal systems, which is a potential focus of future studies.
“

5. The following sentence needs to be clarified: "Gender PD exists if the seriousness
ratings of violence depend on group belongingness of either the experiencer or the
rater".

Our response: This sentence has been simplified to: “Finally, a Gender PD exists if
the seriousness ratings of violence depends on the gender of the experiencers and/or
the raters.” 

6. The authors stated, "…Thus, statements about psychological violence can be
interpreted in more different ways across situations and individuals compared to
physical violence. For instance, sarcasm and irony makes it more difficult for the
receiver to fully comprehend the intention of the communicated statement. This leads
to lower inter-rater reliability in the evaluation of seriousness of psychological
violence".

Please clarify the leap from comprehension of the intention of a communicated
statement to inter-rater reliability.

Our response: These sentences have been clarified and are now written as “Thus,
statements about psychological violence can be interpreted in more different ways
across situations compared to physical violence. For instance, sarcasm and irony
makes it more difficult for the receiver to evaluate the severity of violence in the
communicated statements. This leads to lower agreement between raters' evaluation of
the seriousness of psychological violence.”

7. I had hoped that the distinction among Situational couple violence, intimate
Terrorism, and violent resistance would be included in the analysis. For example,
were the raters informed of the distinctions among these concepts? Did the authors
classify each narrative in these categories in order to determine which categories
of violence were rated as higher in severity, either by the "experiencer" or by the
rater?

Our response: We have added this as a limitation to the current study. The limitation
section now includes the following sentence: “Furthermore, the participants were not
informed, or educated about, various types of violence, for example they were not
introduced to the distinction between situational couple violence, intimate
terrorism, and violent resistance.“. 

8. Was sexual violence rated as physical violence in this study. Research indicates
that sexual violence adds layers of psychological trauma that raters, in this case,
might rate as more severe, especially based on the gender of the
victim/survivor.

Our response: Yes, sexual violence was categorized as physical violence in this
study. This limitation is spelled out explicitly in the limitation section “Future
studies may look into finer distintions, for example, to what extent sexual violence
differs from physical violence. Sexual may violence give rise to other types of
psychological trauma, which in interaction with the gender of victims, may influence
the severity evaluation of the violence.“

9. The authors stated, "First, the difficulty of communicating psychological violence
may lead to a Calibration PD where psychological violence is perceived as less
serious when it is communicated"…

This is an example of where complementary articulation of a study's findings would be
enhanced by adding a qualitative component. For example, by selecting a subsample of
participants for interviews from which the authors would obtain more detail
regarding their experiences of psychological violence.

Our response: We agree that the absence of a qualitative component is a limitation of
the study, however we also think this is a focus of another study. This limitation
has also been added to the limitation section: “The current study uses a
quantitative method where participants use rating scales to evaluate the severity of
violence. Complementary information may be acquired by adding a qualitative
component. For example, participants could be interviewed to obtain details
regarding their experiences of violence, which for example may facilitate
understanding of the different findings related to psychological and physical
violence.”

10. The authors stated,"… psychological violence, being an indirect reinforcer, is
more difficult to communicate than physical violence… It is argued that this occurs
because psychological violence works as an indirect reinforcer, meaning that
psychological violence requires a learning process. This does not apply to physical
violence because it is a direct reinforcer…"

There is a great deal of literature that indicates social learning is involved for
both perpetrators and victims of IPV-

Our response: We are referring to how the actual violence is carried out. We have
clarified the difference by adding the following sentences:

”There are, of course, various forms of social learning involved both in physical
violence and psychological violence, the difference, though, is related to how the
violence is carried out. The effect of a kick or a blow needs very little or no
interpretation, the pain it inflicts is unconditional and direct. The situation,
however, needs to be interpreted, i.e, why the kick was delivered. This is also true
for psychological violence: why, i.e. a threat, is uttered in a given situation
needs to be interpreted. Contrary, though, to a knick or a slap, the statement that
constitutes the spoken threat needs to be understood and interpreted in itself,
before it can be identified as a threat.” 

11. I have no problem with the fact that non-heteronormative persons were excluded
from this study. I am just not sure if the reason provided is sufficient, i.e., "The
screening of heterosexuality was conducted because we were interested in changing
the gender in the collected texts of violence, and we wanted to make sure that this
occurred in close relationships consisting of a man and a woman". Trans women, for
example, have a deeply felt internalized sense of being a woman.

Our response: The motivation of the selecting hetro-sexual couples have been modified
to “The screening of heterosexuality was conducted because we were interested in
focusing the study towards hetrosexual couples, which is the most common sexual
category.”

12. "Upon completion, they received £2.50 for their participation". Please explain
how the authors arrived at this sum as an incentive and provide a reference. Under
limitations, include a discussion of whether or not this small incentive resulted in
the loss of so much data that out of 136 enrolled, the final sample was 71.

Our response: This payment was based on the fact the Prolific Academic recommended a
payment of £7.50 per hour and that the time to conduct the study was estimated to be
20 minutes. As we were following the recommended payment rates, we expected that
dropouts would not depend on the amount of payment. 

13. The authors stated, "The final data comprised of 68 narrations about physical
violence and 68 narrations about psychological violence. These narratives were
collected from a total of 71 narrators, as some of narrators only produced one
narrative and others produced two ".

Were there specific instructions regarding how many narratives were needed or was
there a space limitation?

Our response: It has now been added that “Each participant wrote one narrative about
physical violence and one about psychological violence.”

14. From an ethical perspective, what steps were taken to mitigate participants'
psychological distress related to recall or re-experiencing adverse events such as
physical or psychological violence? Any debriefing? Referral for psychological
support as needed?

Our response: It has now been added that “The participants were debriefed with the
information that they could contact a professional health care given that their
response had evoked negative emotional reactions.” 

15. "In Phase 3: If the participants had not experienced any psychological or
physical violence, they were instructed to describe a situation that was as close to
this violence as possible".

I am not certain that the experience of being slapped, pushed down some stairs,
beaten severely or some other direct act of violence correlates with "a situation
that was as close to violence as possible".

Our response: We have added a sentence motivating why participants “were instructed
to describe a situation that was as close to this violence as possible” stating that
“This was done to facilitate generations of narratives with a low severity of
violence.” 

16. The authors stated, "…This included narrations involving pregnancy (because it
would not make sense to have a story with pregnant men) or when pronouns were
missing (because gender could not be identified in these narratives)…"

I am confused, here. I thought part of the analysis was done by gender. What happened
to narratives for which gender could not be identified? Were they discarded?

Our response: That is correct. Narrations where we could not identify gender in were
excluded, which is clarified in the preceding sentence: “Narrations where it was not
possible to change gender were excluded from the analysis, which included narrations
involving pregnancy (because it would not make sense to have a story with pregnant
men) or when pronouns were missing (because gender could not be identified in these
narratives).”

17. The authors stated, "An analysis of simple main effects showed that whereas there
were no difference in severity ratings for the female narratives in the two
versions, but a significant difference in how the male narratives were rated with
the stories in the gender reversed condition rated as more severe than those in the
original version. This suggests that when a man is exposed to the same physical
violence as a woman, he is considerably less likely to be seen as a victim. This
effect can not fully be explained by the fact that men possess greater physical
strength and are therefore (independent of the evidence) less likely to be
victimised by a woman, because this explanation does not account for the fact that
the same effect is found for psychological violence".

The question which comes to mind is, aren't men often considered psychologically
stronger than women also? Based on neuroimaging studies involving the role of the
amygdala in emotional regulation Please see:

Eippert F, Viet R, Weiskopf N, Birbaumer N, Anders S. Regulation of emotional
responses elicited by threat-related stimuli. Human Brain Mapping. 2007;28:409–423.
[PMC free article] [PubMed]

See also:

McRae, K. et al. The neural bases of distraction and reappraisal. J. Cogn. Neurosci.
22, 248–262 (2010).

Our response: We are not sure what the reviewers point is here regarding that “men
are psychologically stronger than women '' in relation to the cited fMRI studies.
Both the Eippert et al (2007) study and the McRae study only used women as
participants, so it is not possible to draw conclusions regarding gender in these
studies. We are not questioning the idea that there are gender differences on MR
data in relation to emotional stimuli, however, in our view citing fMRI studies in
the context of our manuscript is out of focus. 

18. The authors stated, "One possibility is that psychological violence is harder to
communicate because it is based on secondary reinforcement, which may lead to it
being harder to detect by the receiver of the information" I think if the authors
provide a clearer definition of secondary reinforcement as they are using it, it
would clarify their point. ..

Our response: We have now added a clear definition of secondary reinforcer: “One
possibility is that psychological violence is harder to communicate because it is
based on secondary reinforcement, i.e. a learned stimuli that previously have been
associated with primary reinforcer or a stimulus that satisfies basic survival
instinct such as physical violence. “

19. Under Guidelines, regarding gender, the authors recommend increasing sensitivity
but do not indicate how this would be done. Also, please check grammar and typos in
this section as well as others.

20. I was concerned about the following wording: "If you are male rater, or are given
evaluations of violence from a male rather (Typo, please edit), then consider
increasing your sensitivity, compared to if you are female, or given evaluations
from a female rater.

Again, just recommending that someone increases their sensitivity does not constitute
a fait accompli; it's more likely to evoke cognitive dissonance. Guidelines need to
be accompanied by "how to" instructions.

Our response to 19 and 20: Sensitivity has been defined as “In some contexts
elevators needs to increase their sensitivity, i.e. to say that that a situation is
violent although they evaluate the strength of violence to be less than their
accepted threshold for saying this, and in other contexts the evaluator needs to
decrease their sensitivity, i.e. to say that no violence occurs although they
evaluate strength of violence to be above the threshold:”. Grammars and typos have
been checked.

21. I would omit the entire guidelines from this manuscript, as the intent was not to
write a manual. Stick to the purpose of the study. Your topic and findings are
important enough.

Our response: This section was included following comments from people reading our
manuscript that felt that guidelines were needed to get people to change their
behavior given the results of the study. Therefore we have chosen to keep this
section. However, if the reviewer or the editor repeated the suggested to remove the
guidelines then we are happy to do this. 

22. "Guidelines to victims of violence- If you are communicating psychological
violence, try to be very clear of whether you find this violence severe, or not.
This message may not get across the person you are communicating it to, and it may
also be viewed milder than what you feel that it actually was. If you are male, try
to emphasize the severity of violence more, the violence made to you may not be
taken at face value.

There is so much involved in reporting violence as a victim and so many phases one
must go through (See Transtheoretical model of change and IPV), and once the
decision is made to report IPV, the reporting itself is traumatic; So, the latter
quote can appear as if the authors are engaging in "Victim blaming". I am sure they
are not, but rewording that paragraph is important.

Our response: This paragraph has been rewritten so the elevator, and not the victim,
are viewed as responsible for the poor communication: “People that you communicate
psychological violence to may be poor at understanding the severity of the violence.
Therefore, try to be very clear regarding whether the violence is severe, or not.
Otherwise the severity of the event may not get across to the person you are
communicating it to, and it may also be viewed milder than what it actually was. If
you are male, then the evaluator may view the violence as being less severe than it
actually is, so the violence made to you may not be taken at face value. “

Other items that need to be addressed:

• DOIs missing from references

Our response: the missing DOIs are added

• Lots of grammatical and spelling errors throughout.

Our response: This has been corrected.SFEdit

• Sometimes, no spaces are left after periods

Our response: This has been corrected.SFEdit

• Sometimes, extra spaces are left between paragraphs

Our response: This has been corrected.SFEdit

• Keeping "person" and "voice" consistent in the writing would be helpful. The
authors shift from first person plural to third person- and from active to passive
voice throughout the manuscript.

Our response: This has been corrected. SFEdit

Reviewer #2: 

The authors aimed to examine perceptual discrepancies in narrators’ and receivers’
understanding of the seriousness of IPV. They demonstrate that there are some
differences, and notably gender differences, in attributions of seriousness. This
study is novel and the notion of perceptual differences has implications for both
prevention programs as well as intervention services.

My main concern is the conceptual definition of the main construct, “seriousness.” It
is unclear what this concept means. The authors seem to indicate that it is
synonymous with severity as they use the terms interchangeably throughout. However,
how did the participants interpret the question regarding how “serious” the incident
was? Was there a pilot test with a debriefing session for participants prior to this
study? There are a lot of interpretations of the construct of seriousness that could
pose problems for this study in terms of reliability and what the findings mean. Is
it likelihood of acute physical or psychological trauma? Is it long-term health
effects? Does it tap into the extent to which the behavior is illegal or the extent
to which it should be punished by the criminal justice system? The authors mention
Johnson’s typology, is seriousness a reflection of the extent to which participants
think that the incident narrative is reflective of a relationship marked with
violent coercion compared to deficits in managing anger? With regard to gender
differences, one could respond that violence against women incidents are more
“serious” because they are embedded in larger structural systems that make economic
independence less likely and thus women may be less likely to be able to leave
abusive relationships. In sum, information regarding the validity and reliability of
the single item indicator of seriousness is needed as it could be open to much
interpretation.

Our response: We have responded to the reviewers comments by expanding the relevant
section in the method section to 

“No specific definition of the relevant concepts “physical violence”, “psychological
violence” or “seriousness of violence” were given. This choice was made because our
main focus was to study how the severity of these concepts are communicated, rather
than how the concepts are defined. This allows for an empirically grounded usage of
these concepts, where we can monitor the difference in severity ratings of these
concepts for people experiencing the events related to the concepts and people
receiving text descriptions of the events. To be clear, we understand that concepts
used can be interpreted differently depending on individual differences and
backgrounds of the tested population. For example, the concept “seriousness of
violence” could be interpreted differently depending on whether the participant
emphasizes to what extent it has implication on; emotional suffering, physical
suffering, legal consequences, social consequences, etc on long or short time
scales. Thus, the purpose here was not to provide an exact definition of this
concept, but to study what ratings the concepts evoked in the participants given
that the participants in the phase 1 and 2 were generated from the same population.” 

The authors report that the sample is from the United States. Perceptions of injury
and harm are not only tied to gender, but strongly tied to race. Notably, Black
women are consistently perceived to be less impacted by violence, trauma, and health
issues. For example, their pain is perceived to be less by health practitioners in
hospitals and doctor’s offices. To what extent might these results be influenced by
race? Did the vignettes use any names, real or fake, to communicate the gender of
the narrator and the assailant? Was the distribution of the race of the narrators
and perceivers comparable? It is likely if there were no names used that the
perceivers mentally envisioned a scenario with two people of the same race as
themselves. If the distributions are comparable, I think that the authors can
reasonably conclude that this does not pose an issue, but it should be mentioned in
the methods and discussed in the discussion section.

Our response: We understand that the race of the people in the narrative may
influence the seriousness ratings, and that there also may be an interaction effect
with the race of the evaluators in phase 2. However, although this is an intriguing
research question, it was not the focus of the current study. 

To clarify this, we have added the following sentence In the method section:
“Furthermore, we did not collect data on the race of the participants, nor did we
explicitly ask if the participants were sicgender, because this was not the focus of
the current study. ” 

In the limitation section of the discussion we further spell out that race may be an
important factor, but at the same time we could not analyse based on race because
data on race was not collected, nor was it possible to extract race information from
the narratives: 

“Furthermore, the current study does not investigate the race of the persons
experiencing the violent act and how this interacts with the race of the evaluators.
For example, the seriousness ratings may very well depend on whether the victim is
black and white, and whether the elevator is of the same or a different race.
However, the narrations produced in the current study typically did not include
information about the race of the victim or the offender, nor did we collect data on
the race of the participants of the study, so it was not possible to analyse how
race interacted with the ratings.”

Minor and Miscellaneous

- The authors mainly refer to gender differences, but in the methods they include
discussion of sex ratios. The authors report on sexual orientation, but not gender
identity. Were all participants cisgendered?

Our response: That is the correct, hetrosexuality was an inclusion criteria for the
current study, however, we did not explicitly ask the 

participants if they were sicgender. The latter is clarified in the following
statement:

“Furthermore, we did not collect data on the race of the participants, nor did we
explicitly ask if the participants were sicgender, because this was not the focus of
the current study.” 

- The authors discuss Johnson’s typology. Although they mention violent resistance,
included in his expanded typology is also mutual violent control. Although a minor
omission, it is relevant given the emphasis of this study is on gender parity.

Our response: We have listed mutual violent control in the introduction by adding the
following sentence: “Finally, mutual violent control is when a couple can be
described as “two intimate terrorists battling for control".”

---

## [Decision Letter · Decision Letter 1]

26 Jul 2021

What You Say and What I Hear – Investigating Differences in the Perception of the
Severity of Psychological and Physical Violence in Intimate Partner
Relationships

PONE-D-20-32794R1

Dear Dr. Sikström,

We’re pleased to inform you that your manuscript has been judged scientifically
suitable for publication and will be formally accepted for publication once it meets
all outstanding technical requirements.

Kind regards,

Astrid M. Kamperman

Academic Editor

PLOS ONE

Additional Editor Comments (optional):

Reviewers' comments:

Reviewer's Responses to Questions

**Comments to the Author**

1. If the authors have adequately addressed your comments raised in a previous round
of review and you feel that this manuscript is now acceptable for publication, you
may indicate that here to bypass the “Comments to the Author” section, enter your
conflict of interest statement in the “Confidential to Editor” section, and submit
your "Accept" recommendation.

Reviewer #2: All comments have been addressed

2. Is the manuscript technically sound, and do the data
support the conclusions?

Reviewer #2: (No Response)

3. Has the statistical analysis been performed
appropriately and rigorously? 

Reviewer #2: (No Response)

4. Have the authors made all data underlying the
findings in their manuscript fully available?

Reviewer #2: (No Response)

5. Is the manuscript presented in an intelligible
fashion and written in standard English?

Reviewer #2: (No Response)

6. Review Comments to the Author

Reviewer #2: (No Response)

7. PLOS authors have the option to publish the peer
review history of their article (what does this mean?). If published, this will
include your full peer review and any attached files.

If you choose “no”, your identity will remain anonymous but your review may still be
made public.

**Do you want your identity to be public for this peer review?** For
information about this choice, including consent withdrawal, please see our
Privacy Policy.

Reviewer #2: No

---

## [Editor Report · Acceptance letter]

2 Aug 2021

PONE-D-20-32794R1 

What you say and what I hear – Investigating differences in the perception of the
severity of psychological and physical violence in intimate partner relationships 

Dear Dr. Sikström:

I'm pleased to inform you that your manuscript has been deemed suitable for
publication in PLOS ONE. Congratulations! Your manuscript is now with our production
department. 

Kind regards, 

on behalf of

Dr. Astrid M. Kamperman 

Academic Editor

PLOS ONE